# The Research on Sustainable Tourism in the Light of Its Paradigms

**Małgorzata Kieżel *** , **Paweł Piotrowski *** and **Joanna Wiechoczek**

Department of Marketing Management and Tourism, the University of Economics in Katowice, 40-287 Katowice, Poland; joanna.wiechoczek@ue.katowice.pl

**\*** Correspondence: malgorzata.kiezel@ue.katowice.pl (M.K.); pawel.piotrowski@ue.katowice.pl (P.P.)

**Abstract:** This study investigates the nature and specificity of the concept of sustainable tourism against the background of the paradigm of sustainable development (with regard to the conditions for the implementation of its practices in Polish conditions). The study assumes the hypothesis that researching sustainable tourism as a category within a new scientific concept—i.e., an emerging paradigm of sustainable development—requires the use of an appropriate scientific methodology. This study, in its essential part, has an overview and theoretical character. A critical analysis of the literature on the subject from books and journals, as well as Internet sources, is used in the study. Documentary and observation methods are applied, and the results of qualitative research based on case study research methodology are presented; thus, the empirical part of the paper has an exploratory nature. Research on sustainable tourism as a category within an emerging paradigm of sustainable development shows that researchers use an appropriate scientific methodology, which is compatible with the interpretive paradigm in the highest degree. In case studies, the research is often limited to the analysis of one example. Due to the prevalence of studies applying qualitative methods, an interpretivist approach is prevalent, while a functionalist approach, associated with quantitative research and model testing, is less frequent.

**Keywords:** sustainable development; sustainable tourism; research paradigm; structural functionalism; interpretivism; methodology; methods; scientific method; qualitative research; quantitative research

## 1. Introduction

Due to the complex character of the concept of sustainability and several regional determinants related to the functioning of tourism in Polish conditions, there is a noticeable problem of the measurement and assessment of the phenomena and processes implemented in the context of sustainable tourism. This raises specific questions about methodological studies on sustainable tourism. For this reason, the purpose of this research is to investigate the nature and specificity of the concept of sustainable tourism against the background of the paradigm of sustainable development and to determine the conditions for the implementation of its practices in Polish conditions. It is also important to identify and describe possible research problems in the sphere of sustainable tourism that determine the adoption of a specific methodology within a given type/kind of research paradigm.

This study assumes the hypothesis that researching sustainable tourism as a category within a new scientific concept—i.e., an emerging paradigm of sustainable development—requires the use of an appropriate scientific methodology, which is compatible with the interpretive paradigm in the highest degree. Researchers representing this paradigm rely on a well-established theory and prefer qualitative trends in research methodology. It must be added that there is a dispute between the supporters of qualitative and quantitative methods. This is a consequence of the so-called paradigm conflicts, i.e., ideological disputes concerning the superiority of a given paradigm. These studies

holistically approach phenomena and can be described as heuristic and generating theses. They have an inductive character, which is manifested in deriving general conclusions from the premises that are their particular cases. Qualitative methods (qualitative research) are based on the assumption that in-depth analyses of a smaller number of cases are a better method to study certain problems than superficial studies of a large group of cases. Moreover, they assume that it is better to study many issues through an in-depth understanding of the reality of the representatives of a given community rather than through the use of a previously prepared theoretical model, as is usually applied in quantitative methods. This results from the fact that both in the dimension of spatial and functional analyses (e.g., development of reception areas), and especially in social analyses (ways and forms of implementation of tourism needs), the concept of sustainable tourism should consider the actual attitudes and behaviors of individuals and social groups participating in tourism.

The interpretation of sustainability as a harmonious development has adopted the features of the paradigm that defines the way of thinking about tourism and the functioning of all categories of tourism reception areas. As a result, the concept of sustainable tourism has been adopted as the theoretical basis and the method of interpretation of research in tourism. The concept of sustainable tourism concerns an activity in which there is a balance between the interests of tourists or organizers and the interests of the local community. For this reason, planning and organizing tourism development based on the principles of sustainable development should cover all the above-mentioned aspects [1] Its practical application requires the development and application of many testing, evaluation and measurement methods.

At the same time, an existing discrepancy in the formulation of sustainability objectives in tourism in various perspectives should be noticed [2,3]. From a natural and humanistic perspective, developmental issues are considered through the prism of conditions of tourism development, tourism assets and the socio-economic environment. On the other hand, in an economic approach, the sustainability of tourism development is emphasized in the context of the stability of the entities operating in this industry, as well as the broadly understood tourist attractions, which allows for the implementation of functions towards local communities by tourism. This specific separation of goals in the development of tourism has consequences in the form of difficulties regarding the operational development of theoretical assumptions in the research process or regarding the selection of analytical tools that serve the assessment of the implementation of sustainability principles in tourism [4–8].

For the research process, this represents a need to select one of two perspectives. In one of them, focusing on the assessment of sensitivity and the protection of the features of the natural and socio-cultural environment, against the impact of unfavorable tourism phenomena, has key significance. On the other hand, the other is aimed at researching the optimal use of existing resources in tourism development to achieve the improvement of the level and quality of life of individuals and social groups. The methodological dilemma associated with the determination of methods for measuring and assessing activities and processes that can be perceived as "sustainable" in tourism is additionally intensified by the complex conditions and features of socio-economic development in Poland (similarly to other countries of Central and Eastern Europe). This results from the fact that the sector of tourism services is highly susceptible to the impact of contemporary negative natural, economic and socio-political phenomena, and furthermore, the entities which function in this sector in Poland are struggling with certain burdens of the past (e.g., economic structures that have not been fully developed on a local scale, backwardness in terms of respecting the regulations concerning environmental protection, etc.).

The study analyzes and describes several issues that are most important for the adopted objectives. The first part presents the characteristics of the concept of sustainable development and its impact on changes occurring in tourism, as well as the theory and practical aspects of sustainable tourism, together with an indication of specific determinants of development. The next discussed issues include the problems of the research paradigms and the related methodological foundations of research, with particular emphasis on the specific character of sustainable tourism. Selected case studies (in particular,

from the Polish market) are also presented to illustrate alternative methods and techniques for conducting scientific research in the field of sustainable tourism, while considering various entities and subjects of analyses.

The study, in its essential part, has an overview and theoretical character. A critical analysis of the literature on the subject from books and journals, as well as Internet sources, is used in the study. Documentary and observation methods are applied, and results of qualitative research based on case study research methodology are presented; thus, the empirical research in this study has an exploratory nature.

## 2. Background: Literature Review

### 2.1. The Concept of Sustainable Development and Its Impact on Changes in Tourism

The origin or the concept of sustainable development is associated with the search for a solution to the problem of the impact of contemporary economic development on the weakening of the fundamental attitudes that determine the further persistence of the economy and society [9,10]. This problem has become the source of the critical evaluation of traditional theories of economic growth (especially, the neoclassical theory of growth). Trends in production, consumption, and the scientific and technological progress that contribute to the strengthening of crises related to raw materials and energy, growing environmental degradation and pollution as well as the impoverishment of biodiversity are also assessed negatively. The progressing disintegration of various aspects, which are generally approached separately, that determine growth and development (ecological, economic, scientific, and ethical aspects) is also important in the critical evaluation [11].

The concept of sustainable development, which is an expression of the critical response to the increasing interference in natural and socio-cultural resources, assumes that development should be a method of management in which the exploitation of natural resources does not lead to the degradation of the used land, but at the same time, it enables meeting current and future needs as well as the aspirations of society [12–15].

In the general concept, sustainable development was first defined as "development which meets the needs of the present without compromising the ability of future generations to meet their own needs. [ . . . ] Sustainable development represents a process of change in which the exploitation of resources, the direction of investments, the orientation of technological development and institutional change are all in harmony, and enhance both current and future potential to meet human needs and aspirations" [16]. Ensuring a decent life for present and future generations by meeting basic material needs as well as ensuring conditions for the development of human potential is the main objective of development when perceived in this way.

In this concept, it is vital to assume the prevention of the overexploitation of all elements of the natural environment through the rational use of resources and environmental values and to create conditions that enable the preservation of biological and landscape diversity [17]. Its purpose is to find the optimal balance in the relationships in the "economy–society–environment" intersystem while taking into account the limitations of spatial development and ecological determinants. In the modern approach to this development concept, it is emphasized that it does not deny the possibilities of economic growth; however, the condition is that the rate of the reproduction of environmental resources should not be lower than the rate of the consumption of these resources [18].

In Poland, sustainable development is defined as socio-economic development in which a process of integration of political, economic and social actions takes place along with the maintenance of the natural balance and sustainability of basic natural processes, in order to guarantee the possibility of meeting the basic needs of individual communities or citizens of both present and future generations [19]. Defining the concept of sustainable development has enabled the determination of principles that constitute practical recommendations for the formulation of development goals for societies all over the world. It was stated that the new model of sustainable development should pursue economic

growth, take into account limitations that result from ecological determinants, and strive to improve the quality of people's life [20].

Analyzing the assumptions of the concept of sustainable development, it can be seen that this is close to the model of conduct typical of the holistic paradigm by which contemporary perception was shaped under the influence of the theory of systems. This paradigm assumes an explanation of reality in compliance with nature and reason while taking into account moral standards in human life [21]. Unlike reductionism, holistic approaches study issues in a holistic way, together with the environment in which the studied events occur and analyzed processes take place. According to the paradigm, the elements of the studied whole should be perceived in an integrated, multi-faceted way, and not only with a cause-and-effect approach, whereas the structures and processes should be analyzed both in a statistical as well as a dynamic structure. Furthermore, quantitative and qualitative features should also be considered in the studied object or phenomenon [22].

The concept of sustainable development has come within the sphere of interest of many areas of life and has an interdisciplinary character. This determines the necessity of linking many various research disciplines in their cognitive and application dimensions. The issues of sustainability should be studied with a sectoral approach while considering the subject scope of the concept. It should be emphasized that the equal treatment of the three spheres—i.e., economic, social and ecological—that constitute the sphere of real processes represents the essence of sustainable development. This means that acting at the crossroads of the three spheres is a common aspect of both categories. This convergence means that to develop, tourism should function according to the paradigm of sustainable development [23].

## 2.2. Theory and Practical Aspects of Sustainable Tourism

Investments in the sphere of tourism that were made on a large scale in the second half of the 20th century resulted in significant transformations in the natural—and thus also social—environments in various regions of the world. This became a prerequisite for the emergence of the issue of the limits of growth in tourism, which contributed to the rise and evolution of the concept of sustainable tourism. This aimed to be the response to the need to combine various development goals in the context of a broad approach to the relationship between man and the environment and the social expectations and economic functions of tourism, which are sometimes difficult to reconcile with, or even contradictory to, the expected social and economic functions of tourism [24].

The concept of sustainable development in tourism was formulated during World Conference on Sustainable tourism in April 1995 in Lanzarote [25]. The World Tourism Organisation, United Nations Environment Program, UNESCO and European Union Commission developed The Lanzarote Charter for Sustainable Tourism. It assumed that all tourism activities have to comply with sustainable development and should support social, environmental and economic development [26]. This means that they should aim at [27]:

- full integration with the natural, cultural and social environment,
- cooperation at all levels, from local to national and international in a vertical and horizontal structure,
- improvement of the quality of life of local communities,
- cultural enrichment of every place of tourist destination,
- restoration of environmental balance through assistance in the sphere of technological cooperation and financial aid,
- strengthening and increasing promotion of the system of management of environmentally-friendly tourism,
- implementing actions minimising the negative impact of transport on the environment,
- implementing actions minimising resource-intensity of the tourism sector.

The Federation of National Parks and Nature Reserves defines sustainable tourism as "any form of tourism development, management and tourist activity that supports ecological, social and economic integrity of the areas, while preserving the natural and cultural resources of these areas

unchanged" [28]. It should be emphasized that the concept of sustainable tourism, also referred to as gentle or environmentally friendly tourism, should not be identified with a specific form of tourism, but it should be related to the method of organisation and functioning of tourism economy as a whole [7,29,30].

There are differences in the sphere of approaching sustainable tourism. The trend directly based on the model of sustainable development gives a broad dimension to the concept of sustainable tourism [31]. The trend defines sustainable development as each form of tourism development, tourism management and activity that supports ecological, social and economic integrity of lands as well as preserves natural and cultural resources of these areas in unchanged condition for future generations [32]. Another definition of this trend describes sustainable tourism as a form of tourism development or tourist activity that respects the environment, ensures long-term protection of natural and cultural resources and is socially as well as economically acceptable and fair [33].

An approach identifying sustainable development with eco-tourism, alternative or green tourism can also be found in the literature [34–36]. In this context, there are two approaches. One of them, the so-called approach of a "small scale" identifies sustainable tourism with a form of tourist traffic, whereas the other approach subordinates all forms of business activity to the principles of ecological ethics [37]. According to the assumptions of sustainable development limiting tourism to a specific form of tourism or complete subordination to ecological ethics leads to distortion of the general idea of the paradigm of sustainable development. According to its overriding principle, sustainable development comprises activities ecologically admissible, economically justified and socially desirable [38]. Therefore, sustainable tourism should concern the tourism supply side (for example ecologisation of tourist regions, tourist products, tourist staff) and the demand side (for example ecologisation of tourist consumption, i.e., shaping pro-ecological needs and consumer behaviours) [31].

The idea of linking the natural, social and economic order in the socio-economic development, which is exhibited in the theory of sustainable development, is reflected in the holistic approach to tourism phenomena in the concept of sustainable tourism. Scientific discussion on relationships between natural, cultural, social and economic values in tourism development is conducted on the basis of the assumptions of the concept of sustainable development [30,39–45].

Sustainable tourism is the result of the study of relationships between tourism, environment and development, perceived as a tool of implementation of sustainable development, but also as a tool for the development of tourism itself [46]. The need to distinguish the very idea of sustainable tourism and the perception of tourism as a factor of sustainability in socio-economic development should be pointed out at this point because they concern different problems. In the first case, we deal with seeking the way of reaching balance in the broadly perceived tourism (considering, for example, the forms of tourism movement, tourist facilities, transport, the functioning of very diverse reception areas). In the second approach, tourism activity and its economic effects are used to implement the principles of sustainability in local development, for the benefit of social and economic development. As a result, these significant differences mean that both approaches require individual solutions on the theoretical as well as the methodological level [47].

## 3. Determinants of Sustainable Tourism Development

The issues of tourism development in Poland in a sustainable approach are mainly determined by political and legal determinants of international and national dimension. General the socio-economic program of the European Union expressed in the Europe 2020 Strategy as well as "Strategy of innovativeness and efficiency of the economy" developed on its basis are the key documents in this area (besides other international documents adopted among others by UN, World Travel and Tourism Council and World Travel Organisation) [48]. The Environmental Protection Law, Dz. U. (Journal of Laws) no. 62 of 2001, item 627 as amended is another important document taking into account environmental issues in Poland, whereas Charter for Sustainable Tourism (1995) strictly refers to sustainable tourism.

Entities preparing and offering tourist products (e.g., hotels, tourist and catering companies) as well as those shaping tourist, recreation and transport infrastructure, investing in creation and maintenance of various forms of natural environment protection, etc. (e.g., local governments, managing national parks, museum and historical premises or investors) are another vital determinants (of supply character) affecting implementation of the principles of sustainable tourism and the pace of its development [49]. Now, they are hotels that most try to implement solutions that fit in the concept of sustainable development. For example, Accor Group hotels, also operating in Poland, joined PLANET 21 project that considers the principles of sustainable construction, ecological room/premise supply, creation of diverse offer, as well as environmentally-friendly energy, water and waste management [50]. Managing a hotel in compliance with these principles allows not only for preserving the environment in the natural condition, bur firstly ensures financial savings in long-term perspective resulting from the reduction of operating costs.

Behaviours of consumers of tourism services that have a considerable impact on the development of sustainable tourism are the third determinant (of the demand character) comprised in this case study. Special attention is focused on them since the growing number of buyers seeking tourist offers shaped in compliance with the principles of sustainable development can encourage the entities on the supply side to create them faster and in a more diversified way.

Over several recent years, multidimensional transformations occurring in these behaviours have been observed. They result from many factors of diverse character. The most important include polarisation of consumer incomes leading to growing disparity in the level of their wealth; intensification of population migration (especially economic migration but also political); mitigation or removal of barriers in crossing state borders; increasing life expectancy and population ageing (especially in developed countries); growing pluralisation of forms of social life; intensifying society digitisation under the impact of popularisation of new technologies in private and business life (leading to permeation of work and holiday); growing pace of contemporary life; as well as growing ecological, health and social awareness (also under the impact of noticing greater degradation of natural environment and occurrence of problems of social character by a growing group of consumers). The emergence of new needs in the sphere of tourism or other ways in which existing needs are met is the result of the influence of the above-mentioned factors on consumers. This brings multidirectional changes in behaviours of consumers of tourist services. The most important are shown in Table 1.

Analysing the trends of changes in consumer behaviours in the sphere of tourism, it can be stated that while being more and more mobile and networked [51–53], contemporary tourists seek more and more often unique, mainly personalised values (for example they willingly get to know new places as well as regional and local cultures, discover authentic areas, gain new valuable experiences while communing with real nature and/or establishing contacts with local communities, etc. and prefer pro-ecological consumption) and do smart shopping which is reflected for example in consuming products together with other buyers. Therefore, we can speak about the rise of the so-called tourist-trysumer (from trying consumer). The change of both the travel theme as well as ways, places and forms in which they relax and spend time is the consequence of the evolution of their behaviours. The increase in the number of tourists characterised by the above features constitutes an important determinant of the development of sustainable tourism.

The trend which is disadvantageous for sustainable tourism, i.e., a growing share of Polish tourists in mass tourism, should also be mentioned here. It is proved among other factors by growing number of visitors to most popular destinations (mainly by the seaside and in the mountains) and tourist facilities during holidays [54]. Tourists' interest in accommodation facilities of a high standard is also growing. It indicates an increasingly greater need to relax in comfortable conditions. This results in a negative impact on the natural and cultural environment. On the other hand, on the supply side, it is also reflected in the loss of regionalism in the architecture of newly constructed accommodation and catering facilities, exclusion of local dishes from the menu, the loss of the character of local culture in the sphere of entertainment, commercialisation of souvenirs, etc.

**Table 1.** Most important changes occurring in consumer behaviours that determine the development of sustainable tourism.

| Trend of changes | Essence |
|---|---|
| Individualisation of needs and expectations of tourists (related among others to their self-fulfillment) as a result of consumption heterogenization | emphasis by tourists of their uniqueness and individuality, and as a result, seeking highly personalised, unique offer |
| Growing tourists' demands in the sphere of quality, safety, etc. | seeking authentic (brand), trustworthy products by tourists, focusing on the comfort of travel and rest, as well as the quality of the service |
| the birth of a tourist seeking authenticity, new cultures and exciting experiences | tourists' wish to get to know other regions, cultures, seeking unique experiences, the wish of experiencing while travelling |
| Growing spatial mobility of tourists and their growing "networking" | quitting a settled life related to the realisation of aspirations among others, regular use of new technologies in private and business life (effective functioning in real and virtual environment) |
| Growing awareness and sensitisation of tourists towards health, ecological and social issues | tourists' pursuit of healthy activation of lifestyle and growing wish to live in harmony with nature (under the impact of the adoption of LOHAS (Lifestyle of Health and Sustainability) lifestyle among others) promotion of attitudes of respect towards environment among tourists (through social networking media among others) |
| Co-consumption of products by tourists (including growing interest in "couchsurfing" and "room-sharing") | rational consumption, including shared use (lending, renting, exchanging, etc.) by consumers/tourists of products and services (including organised forms of sharing products) |

Source: own case study.

Referring to the features of modern tourists, results of regular research concerning ecological awareness and behaviours of the residents of Poland, conducted at the order of the Ministry of the Environment in the period of August–September 2018 should be mentioned [55]. They showed that Poles increasingly more often believe that environmental protection can positively affect the development of the state economy (around 84%). As regards natural environment, they firstly notice the problems associated with air pollution (especially by industrial plants), waste (around 60% of respondents declared that they segregate the rubbish) and climate change (almost half of respondents showed the necessity of possibly fast reduction of greenhouse gases in the interest of future generations). The respondents want to protect the environment mainly because of the concern for health and future generations (whereas half of the respondents believe that each citizen is responsible for the condition of the natural environment, according to 35% respondents the situation of the environment depends on the activity of local authorities and according to 32% on legal regulations), and also care about nature as the value in itself. It can be noticed that the attitudes of Poles more and more correspond with the assumptions of sustainable development, which could also be positively used for faster development of sustainable tourism. "Circular economy" and corporate social responsibility proved to be the least known among Polish people (60% of respondents do not understand these notions). This can result from the fact that this type of activities is primarily implemented by business entities (in the case of tourism, for example, hotels, travel agencies, restaurants) or institutions of various type, and not directly by the very consumers. According to around 1/3 studied Poles, shaping ecological behaviours among the society should be the task of mainly local governments and voivodeships, as well as the government and central authorities. This conviction of the respondents seems vital for consumer education, i.e., creation and development of tourists' attitudes consistent with the assumptions of sustainable tourism in this case.

Summing up the above considerations it should be stated that sustainable tourism should be reflected in such form of tourist activity that takes place with environmental and cultural awareness and respect, while securing long-term preservation of its values at the same time. It should be recognised that it is a compromise between the competitive interests of the tourism economy and the needs of

ecosystems [56]. It is the compromise pursued on the basis of optimality, rationality and efficiency criteria. It is determined by various alternative possibilities of use of limited production resources that have unequal productivity. Alternative character concerns the selection of destinations, directions and types of tourist activity undertaken. Its limited character refers to the environment potential and its elements, the quality of which determines at the same time the process and efficiency of tourism development. The limited nature of environment potential in conditions of demographic growth and development of higher and higher economic forms, as well as pursuit of various needs on the borderline of biological and psychological needs, make the environment, its resources and values an economic good of public character and thus it must be protected.

## 4. Materials and Methods

### 4.1. The Issues of Research Paradigms and Methodological Basis of Research

Science is defined as a specialised cognitive activity performed by scholars, targeted at objective recognition and understanding of natural, social and economic reality, and creation of the premises for the use of acquired knowledge for the purpose of transforming the reality according to human needs [57–59] divided sciences into nomological, that aim at determining the laws of development of fragments of reality, and typological sciences, aiming at determining types of things and types of phenomena occurring in studied fragments of reality. This division corresponds to another division into nomothetic sciences that try to explain phenomena through the formation of laws and theories and ideographic sciences that describe phenomena and processes while not aiming to explain them. According to the other division, the reality of economic sciences is even more complicated. Some disciplines of this science area are included in nomothetic sciences (descriptive knowledge), identifying laws and regularities (theory of economics, descriptive economics and its sub-disciplines), whereas others in ideographic sciences (e.g., sciences of management, marketing), that primarily comprise normative knowledge.

The former division included economic sciences among typological sciences. They have specific features, differentiating them from others and affecting the specific character of methodology. They comprise complexity of studied phenomena that are associated with the so-called syndromaticity—linking a large number of features and relationships of phenomena in such a way that they have to be approached holistically because none of them can be omitted. Another feature is a difficulty, and consequently little application, of precise methods and sometimes even their omission. This specifically concerns laboratory experiments that are expensive and for which it is difficult to separate economic phenomena in such a way that they should proceed in artificial conditions just like in a normal situation. Due to this, less precise methods are more frequently applied, e.g., natural experiments, observations, surveys or tests. Using colloquial language is the third feature of economic sciences because the majority of economic phenomena is widely-known to people and they have colloquial names. As a result, this can limit the preciseness of economic sciences. The fourth feature, incomplete objectivity can result from the need to evaluate the studied phenomena by researchers or from the researcher's attitude (ideology, own interest) [60].

Scientific knowledge comprises observable phenomena and thinking processes. It provides and justifies mechanisms, causes and their results in theoretical and practical approach while applying research methods, techniques and tools that allow for verifying the correctness of scientific assumptions and obtained research results [61]. Five functions of science can be distinguished here [62]:

- descriptive, associated with describing,
- explanatory, representing explaining,
- predictive, responsible for forecasting,
- designing, that enables developing,
- evaluation that is associated with verifying.

All the above functions are vital and there is a feedback between them. Describing is always the beginning of the process. It supports explaining and forecasting, which in turn enables designing new solutions. Explaining is a form of reasoning. Its essence is seeking the premises justifying the state of things that occurred due to specific reasons. If the problem is superficially described, with no appropriate explanation, forecasting will not be appropriate. On the other hand, if forecasting is not correct, the new thinking construct will be wrong. Creative activity has a scientific character if it is conducted with the use of canons of science as well as suitable and appropriate research methods.

Canons of science are a set of principles of fundamental importance for scientific recognition and implementation of scientific research. Fundamental scientific canons include among other paradigms of science that perform a vital role in research on new areas. The paradigm in the approach introduced by Kuhn is a set of notions and theories constituting the fundaments of a given science. He distinguished its broad and narrow understanding. In a broad approach, a paradigm is a specific type of matrix orienting the attitudes of everyone pursuing a given field of knowledge [63].

Lakatos [64] introduced the notion of the so-called research program that forms a set of basic theories called paradigms. Specific theories consistent with experimental data that specific science investigates are formulated on their basis. The paradigm of a scientific method that constitutes a criterion of recognition of an activity as scientific is the most general paradigm. Specific theories are tested and built according to the principles identified by Kant and Popper [65], so they are usually either verifiable or falsifiable. According to Lakatos's model, science is a correctly constructed paradigm and specific theories experimentally verified [66].

A good paradigm should be characterised by several features. It must be logically and conceptually consistent, be as simple as possible and contain only the concepts and theories that are necessary for a given science. It must also allow creating detailed theories consistent with known facts. This means that it should be possible to build on its basis, falsifiable and verifiable theories that would well explain well-known empirical facts [67,68].

However, the concept of falsifiability as a distinction between science and non-science faces problems within some sciences, in which it is not always appropriate to rely on a methodology that is considered to be mathematically or statistically measurable. For example, in the humanities, although there is a possibility of falsification based on collected historical material, for some researchers, such as logical positivists, this kind of knowledge can only be perceived as a systematic description of a phenomenon that will never be fully developed in terms of science. Similar problems also arise in many psychological and sociological theories, although some researchers try to recognize this kind of knowledge as science, because of the nature of its creation that consists in researching, discussing and publishing the results, however without the possibility of their falsification. This also refers to the concept of sustainable development that changes the point from which the subject of scientific research is viewed in terms of the assumptions of economics, including the neoclassical one, where all theoretical considerations and empirical studies were analysed through supply and demand [69].

Paradigms approached as fundamental scientific assumptions in terms of the nature of reality that is taken for granted and not negated in research, can be classified in research while taking into account three major criteria of for differentiation, i.e., variability or stability of the social world, objective or subjective nature of reality (reality as a product of mind to some extent) and the necessity to participate in a given process for its recognition, or lack of this necessity. A comparison of these criteria (in the latter two cases in a linked form) results in the identification of four basic paradigms, i.e., functionalism (rationalist paradigm), structuralism, interpretivism and radical humanism [67] (Tables 2 and 3).

The selection of a research paradigm has a large impact on the choice of research methods and has a clear influence on preferences regarding the use of quantitative and qualitative methods in research. This results from the fact that they group different schools and research trends that share common ontological, epistemological assumptions as well as research preferences [70,71].

**Table 2.** Paradigms in Morgan's and Burrell's approach.

| Criteria | The social World is Changing | The Social World Is Continuously the Same |
|---|---|---|
| The world has an objective character, the researcher can analyse it from the outside while using abstract theoretical models | Radical structuralism | Functionalism |
| The world has a subjective character, its understanding demands to go into its interior | Radical humanism | Interpretivism |

Source: adapted from [67].

**Table 3.** General characteristic of paradigms.

| Paradigm | Main Idea | Subject of Research | Knowledge Specific |
|---|---|---|---|
| **Functionalism (rationalistic paradigm)** | The studied community is perceived as a system of mutually related elements that perform functions for the balance of the whole. It explains phenomena in historical and psychological categories. | A researcher analyses collectivity with the use of the theoretical model developed before. | Knowledge as an object that can exist independently of human actions and perceptions; there is a consensus about the value of knowledge and work. |
| **Radical structuralism** | It is a type of structuralism, where the structure is more important than acting, the idea that besides often changeable expressions of reality, there are structures constituting its fundament. This paradigm represents the assumption of the presence of structures "hidden" under the studied surface of phenomena and non-direct cause-and-effect relationships. | Specific models of relationships called structures are the actual subject of structural research. Specific criteria of structure: it has the character of an orderly system (it consists of such elements that modification of one of them causes modification of others); it can be predicted how a given structure will react if one of the elements is modified; the structure always belongs to a group of structures of similar relationships. Researchers adopting this paradigm do not apply in field research the previously developed or borrowed theoretical model. They try to apply the so-called thick description that considers not only behaviour but also its context. The research objective is to understand what is important for the members of a given community because they create norms and values and authentically understand the system in which they act (the researchers should not arbitrarily attribute functional meaning to what they observe). | Knowledge as an object that can exist independently of human actions and perception; the value of knowledge and work is refuted, and it becomes a source of conflict. |
| **Interpretive** | Its representatives assume that to understand the rules of a given phenomenon, it is necessary to penetrate the reality of people who use it every day. Paradigm developed in opposition to functionalism. | | Knowledge is a social practice of knowing. There is a consensus about the value of knowledge and work. |
| **Radical humanism** | Similarly to interpretative paradigm is based on the conviction that society is not a real being; however, it perceives the role of science in a different way. Reality is socially created and sustained, although the assumption is connected to the whereby actors are seen as prisoners of their existence. | Social reality is socially created but it happens at the expense of people who grow "trapped" in the unreal world, even though recognised as necessary. | Knowledge as a social practice of knowing; The value of knowledge and work is refuted, and it becomes a source of conflict. |

Source: adapted from [67,70,72,73].

Moving on to methodology of sciences, in a narrow approach it is a science of research methods, i.e., of creation of research activity structures and their application to recognition of things or phenomena of a given type. On the other hand, in a broad meaning, it is a science concerning not only methods, but circumstances of research (social, psychiatric and technical) and features of knowledge. In the first case, it is described as pragmatic knowledge because it consists of statements concerning real facts. On the other hand, there is knowledge about features of scientific knowledge—anti-pragmatic, consists of meta-scientific statements, concerning other knowledge. Components of scientific methodology are divided into descriptive (stating) and normative, which is the basis for further related divisions (Table 4).

**Table 4.** Classification of elements of science methodology (knowledge-creating statements).

| Statements | Descriptive | Normative |
| --- | --- | --- |
| Pragmatic | Information about what methods in a given science are applied for solving specific problems | Methodological postulates showing what methods should be applied for solving problems of a given type |
| Anti-pragmatic | They inform the features of works published as scientific | They show the features (shared and unique) of research results to recognise them as scientific knowledge |

Source: adapted from [74,75].

In this case study, in the context of methodology of sciences, pragmatic statements are focused on. Two groups of elements are included. The first, descriptive, provide knowledge on what methods in a given science are applied for solving given problems. The other group of elements, i.e., normative, are created by methodological postulates that show what methods should be applied for solving problems of a given type.

*4.2. Methodological Approaches in Scientific Research in the Sphere of Sustainable Tourism—Case Study Research*

The character of information acquired as a result of measurement of primary sources allows for distinguishing quantitative research, where market facts are registered and on the basis of which statistical image of a phenomenon and market processes take place, and qualitative research that supports recognition of opinions, preferences and attitudes towards market phenomena and processes [76].

Quantitative studies are objective, and they apply standardised tools that are statistically verified. Their basis is formed by hypothetical and deductive sciences consisting in proposing theses and formulation of theories, as well as reasoning consistent with the trend of logical result. In this type of research, researchers define some categories and expectations a priori before they start research, and then they find relationships between them [77]. The research is usually based on questionnaire surveys conducted among relatively big samples of respondents, most often representative for purposeful population, with the use of statistical and mathematical methods at the selection of sample and compilation of results. It consists in collecting data, gained from respondents with the use of forms—formalised questionnaires, surveys and interviews. In this way, based on collected data it can be stated how often and how much various opinions and facts occur in a given community [78]. Quantitative data serve mainly the measurement of the scope of studied phenomena. They are useful while exploring such issues as determination of the market size, recognition of competitive environment, market segmentation, the study of market perception against competitive brands and study of consumer attitudes towards the brand.

Qualitative methods (qualitative research) are based on the assumption that in-depth analyses of a fewer number of cases are better to study some problems than superficial studies of a large group of cases. Qualitative research produces information on the particular cases studied, and any

more general conclusions are only hypotheses [79]. Moreover, they assume that it is better to study many issues through an in-depth understanding of the reality among the representatives of a given community rather than through the use of a previously prepared theoretical model, as is usually applied in quantitative methods. Researchers are more interested here in the answer to the question "why" rather than "how many". This represents the necessity to perform field studies with the use of such methods as interviews—especially uncategorised, participating observations, heuristic methods and shadowing [80]. The so-called projection techniques that enable disclosure of unconscious or hidden views and emotions associated with the studied problem are also an element of qualitative research [81,82].

Google Scholar, ScienceDirect and Scopus databases were searched to identify case studies referring to research associated with sustainable development in Poland. To find them, the following key words were used in the whole texts: Turystyka Zrównoważona and Studium Przypadku (in Polish), and Sustainable Tourism, Case Study and Poland (in English) (Table 5).

**Table 5.** The number of publications concerning the issues of sustainable development identified in databases.

| Language of Publication | Google Scholar | Scopus | Science Direct (Research Articles only) |
|:---:|:---:|:---:|:---:|
| Polish | 59 (7) | 0 | 0 |
| English | 4210 | 2 (0) | 59 (8) |

Source: own study.

Due to a big number of results in Google Scholar database in English (4210 publications), in further analyses, in this case, the search was limited only to results in Polish. In the further stage, papers that were not published scientific papers (Google Scholar) and/or textbooks were rejected. Such papers that in the abstract content did not refer to the issues of sustainable development and/or if there was no access to their full content were also rejected. Analysing titles and abstracts, the focus was on the identification of such works that concern sustainable tourism and not, for example, sustainable transport, logistics, etc., even though with no doubts they are a vital element supporting sustainable tourism. At the same time, case studies in which the subject of analysis is located in Poland were looked for. As a result, 15 works that met the above criteria could be identified.

## 5. Research Results

Identified case studies were analysed from the point of view of the research subject and entity, applied research method and the sample size, the paradigm the research fits in, as well as the function it performs (Table 6). Identified works can be divided into those that concern only and exclusively Poland, and those in which examples are from Poland or where Poland is one of the analysed cases. At this research stage, the empirical part has exploratory nature. Exploratory research intends merely to explore the research questions. This type of research design leaving space for further researches because they usually conduct to study a problem that has not been clearly defined yet. They do not offer conclusive solutions to existing problems.

**Table 6.** Case studies found in the literature that concern sustainable tourism in Poland.

| No. | Author/ Publication | Research Entities | Research Subject | Research Method | Sample Size | Paradigm | Function |
|-----|---------------------|-------------------|------------------|-----------------|-------------|----------|----------|
| 1. | Bohdanowicz [83] | Hotels in Poland and Sweden | The influence of the geopolitical, economic and socio-cultural context of a country on the environmental attitudes and pro-ecological initiatives incorporated in the hotel sector | Quantitative research based on e-mail questionnaire | 349 hotels | interpretivist | descriptive, explanatory |
| 2. | Hendel [84] | Clusters | The use of cluster as an instrument of sustainable tourism implementation in the reception areas | Qualitative research based on the literature and documents | Cluster Beskidzka 5 operating in the area of 5 municipalities | interpretivist | descriptive |
| 3. | Kapera [85] | Hotels | Analysis of present situation, opportunities and barriers in implementation of the concept of sustainable development in hotel industry practices in Poland | Qualitative research based on the analysis of the literature on the subject | - | interpretivist | descriptive |
| 4. | Kapera [86] | Local governments in Poland | Programs of local governments in the sphere of sustainable tourism | Quantitative research based on paper questionnaire | 600 representatives of local governments | functionalist | explanatory |
| 5. | Luc, Tejwan-Bopp, Bopp and Szmańda [87] | Tourist portal | Presenting ETNOS (E-Tourism Native Open Service) tourist portal as a tool supporting the development of native tourism that fits in the concept of sustainable tourism | Qualitative research, the case study of the Internet portal. | 1 portal | interpretivist | descriptive designing |
| 6. | Łącka [88] | Social enterprises | The use of eco-tourism by the entities of social economics for the implementation of their tasks | Qualitative research based on analysis of documents | 1 social enterprise | interpretivist | descriptive |
| 7. | Myga-Piątek and Jankowski [89] | National parks | Presentation of positive and negative examples of the impact of tourist traffic on the natural environment and landscape | Qualitative research—observations, own experiences | 2 national parks: Bieszczady and Karkonosze | interpretivist | descriptive |
| 8. | Niezgoda and Markiewicz [90] | Hotel | Showing consistency between business tourism development and the concept of sustainable development | Qualitative research, a direct diagnostic survey with the use of personal interview and questionnaire | 1 hotel | interpretivist | explanatory |
| 9. | Panfiluk [91] | Participants in events | Examination of the effects of special events organized in Poland in the Podlaskie region, | Quantitative research based on the results of the survey conducted among participants in events | 471 participants in 3 events | functionalist | explanatory |

**Table 6.** *Cont.*

| No. | Author/ Publication | Research Entities | Research Subject | Research Method | Sample Size | Paradigm | Function |
|---|---|---|---|---|---|---|---|
| 10. | Paramati, Shahbaz and Alam [92] | Countries of Eastern and Western part of the European Union, Poland was among them | Analysis of the impact of tourism on economic growth and $CO_2$ emission | Quantitative research based on data of World Bank of 1991–2013 | 28 states of the European Union | functionalist | explanatory |
| 11. | Perano, Abbate, La Rocca and Casali [93] | Small and medium-sized enterprises located in places belonging to Citta Slow | Relationships between territorial certifications and SMEs value creation | Quantitative research | 1148 enterprises from 15 countries, including Poland | functionalist | explanatory |
| 12. | Rohrscheidt [94] | Various entities of the tourism sector (travel agencies, tourism organisations) | Identification of examples of products consistent with the concept of sustainable tourism | Qualitative research | 14 product examples | interpretivist | descriptive |
| 13. | Schliep and Stoll-Kleemann [95] | MAB biosphere nature reserves | To test the effectiveness of the MAB vision and business plan at the local level. | Qualitative research-qualitative interviews with stakeholders and relevant local actors | 3 nature reserves in Poland, Hungary and Czech Republic | interpretivist | descriptive |
| 14. | Świstak, Świątkowska and Stangierska [96] | Hotel chain | Areas of strategies implementing the assumptions of sustainable development on the example of the hotel chain. | Qualitative research including a case study based on the analysis of a strategic document | Accor group | interpretivist | descriptive |
| 15. | Wilkońska [97] | Krakow city | Reflection concerning the possibility of development of big cities towards slow tourist, especially in the context of separated space of selected district. | Research? Qualitative, concerning possibilities of the implementation of the content of the city strategy, referring to the possibility of development of a district supporting low tourism movement | 1 city | | |

Source: own case study.

Analysis of the research entities presented in Table 4 proves that they are highly diversified. However, studies conducted among private entities (eight cases), especially hotels, are predominant. The entities of the public sector, states, local government or national parks are less frequent. Only once they were tourists/visitors. This is compliant with the fact that sustainable tourism is perceived as the concept of management of destinations or entities in the tourism sector. In the case of the research subject, the situation seems even more diversified. What generally characterises analysed works is the fragmentary approach to the issues of sustainable tourism. However, it is not surprising if we consider the fact of the limited length of scientific papers in comparison with the complex character of the subject area. They concern either selected entities, defined spheres of sustainable tourism concept (e.g., only the environmental aspect) or specific spheres of the functioning of a given entity, for example only and exclusively their strategic documents without analysing real actions. As it could be supposed, analysing the used research method, a qualitative approach was prevailing. However, it must be stated that often in the qualitative approach, the studies were based not on primary sources—surveys and interviews, but on the analysis of the literature on the subject, or strategic documents. The sample size depended on research methodology. In case studies it is often limited to the analysis of one example. Due to the prevalence of studies applying qualitative methods, interpretivist approach is prevailing, while the functionalist approach, associated with quantitative research and model testing is less frequent. Consequently, the descriptive function is predominant whereas explanatory character is less frequent.

## 6. Discussion

Starting deliberations concerning the issues of the possibility of scientific recognition, especially research methods for identification and explanation of the problem of sustainable tourism, its nature and possibility of its development and evaluation is an important aspect in the study. This represents taking into account the ontological aspect (that inquires whether the analysis of existence should precede the analysis of recognition), as well as epistemological one (comprising the problem of appropriate path of recognition).

Conducting tourism activity has a very complex nature because it takes place both in geographical space (physical and anthropogenic), as well as in economic, social and cultural space. In an individual dimension, tourism also gains the psychological aspect, because it represents some symbolic and emotional value of a high level of individualization [98]. Consequently, tourism is becoming the subject of interest of researchers who represent various fields and apply diverse methods. This multi-faceted nature of the phenomena occurring in tourism raises the need to approach tourist activity as conducted in the space linking both material and non-material aspects.

The case study adopts the initial ontological assumption that shows that studied aspects of sustainable tourism should be approached as a part of a higher level space (geographical, socio-economic and cultural), rather than as a separate entity. These ontological assumptions affect epistemological aspects and adoption of some assumptions concerning the methods of sustainable tourism research. However, it is assumed that it should be studied in a multi-faceted way while taking into account its specific and complex character, and also considering strong relationships with the external environment. Analysing the trends of further research it seems useful to verify how much the defined features distinguishing sustainable tourism from the higher level space, enable its measurement or description in the sphere of adopted research aspects, depending on the assumed methodological approach, i.e., quantitative, qualitative or mixed.

It should be stated that it is difficult to assess these two types of research and recognise one of them as better than the other. This results from the fact that each of them enables gaining another set of information, that if it is used differently, it supports solving the research problem. Consequently, this causes quantitative research to occur together with qualitative while complementing each other, or possibly it precedes qualitative research while enabling primary recognition of the formulated problem. The purpose of triangulation of research methods may be particularly useful to increase

the credibility and validity of the results and gives a more detailed and balanced picture of the situation. It is an "attempt to explain more fully, the richness and complexity of human behavior by studying it from more than one standpoint [99,100]. "The concept of triangulation is borrowed from navigational and land surveying techniques that determine a single point in space with the convergence of measurements taken from two other distinct points" [101]. It is an appropriate strategy of founding the credibility of qualitative analyses and it becomes an alternative to traditional criteria like reliability and validity [102].

According to Spencer's views, development, on the one hand, consists in a growing diversification of phenomena, and on the other hand in their growing integration, ordering and balancing. For the assessment of the development of sustainable tourism, this represents the need to identify and measure the broadly perceived benefits against broadly approached costs of tourism development. An attempt to consider all or at least the majority of elements of contemporary tourist space would demand application of the research approach of a synthetic and systemic character. Such an approach would be especially important at identification and analysis of the level of tourism sustainability.

For the holistic approach to the issues of sustainable development, it is necessary to evaluate all tourism stakeholders in a given area. This reflects the necessity to consider the opinions of tourists, as well as residents and other entities (in visited places) directly or indirectly involved in tourism development. This results from the fact that both tourists, as well as the local community, can have diverse expectations, but they also bear consequences (positive and negative) in terms of the broadly perceived natural, socio-cultural and economic environment [103].

The specified division into more uniform sub-groups supports the performance of research on the evaluation of the level of sustainable tourism development as perceived by many groups of stakeholders. As a result, subgroups of diverse socio-demographic features, but also different reasons for travelling and related different expectations or travel frequency can be indicated within the group of tourists [104]. On the other hand, within the local community one can distinguish tourist entrepreneurs, workers, employers hired for tourist services, representatives of public administration, representatives of social organisations associated with tourism development among others. However, people not involved in providing services to tourists or people occupationally not active that are affected by tourism development should also be included here.

## 7. Conclusions

Sustainable development as an expanding scientific concept creates its theory, terminology and methodology. This is because describing the phenomena in terms of that concept, and related concepts, requires a proper scientific "basis" that is consistent with accepted canons. This problem is also observed in the sphere of research on sustainable tourism, which is developing as the field/sectoral implementation of the concept of sustainable development that is more and more theoretically advanced. This results from the fact that the paradigm adopted in it requires a new approach to the forms and trends of tourism development that operates on the border of the social, economic and environmental system.

The study indicates the nature and specificity of the concept of sustainable tourism against the background of the paradigm of sustainable development (have regard to the conditions for the implementation of its assumptions in Polish conditions). This theoretical study has a more universal character. Data analyses and case study research identify and describe the adoption of a specific methodology within a given type/kind of research paradigm. Research on sustainable tourism as a category within an emerging paradigm of sustainable development shows that researchers use an appropriate scientific methodology, which is compatible with the interpretive paradigm in the highest degree.

Researchers representing this paradigm rely on a well-established theory and prefer the qualitative trend in research methodology. They often apply the grounded theory, based on the assumption that reality is best understood by actors involved in it. Therefore, the traditional functionalist approach is

rejected here while assuming that it only causes self-confirmation of a given theory (researchers are confirmed in their idea because they find what they seek). Therefore, in grounded theory, the researcher operates in the field with no pre-conceptualised theory. The theory emerges from the very research, it "grounds by itself" in the field during collection of research material in the successive studies, i.e., interviews, observations and text analyses. In this way, the theory emerges that concerns only this specific collectivity but fits it much better. Due to this approach, the grounded theory is based on a coherent system of qualitative methods and sometimes is even identified with them, which is the reason for criticism from scientists exercising qualitative methods, but who use of other approaches (such as organisation ethnography, storytelling). Although in many prestigious universities, representatives of grounded theory are in visible positions (Harvard University or University of California Berkley among others), in Poland this method is not fully recognised yet and raises many disputes.

**Author Contributions:** M.K., P.P., and J.W. participated in all the phases and contributed equally to this work, including the establishment and application of the framework, as well as the writing on paper. In particular: conceptualization, Małgorzata Kieżel, Paweł Piotrowski and Joanna Wiechoczek; formal analysis, Joanna Wiechoczek; methodology, Małgorzata Kieżel and Paweł Piotrowski; results, Małgorzata Kieżel, Paweł Piotrowski and Joanna Wiechoczek; writing—original draft, Małgorzata Kieżel, Paweł Piotrowski and Joanna Wiechoczek; writing—review and editing, Małgorzata Kieżel, Paweł Piotrowski and Joanna Wiechoczek.

**Funding:** This research received no external funding.

**Conflicts of Interest:** The authors declare no conflict of interest.

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
