# Peer review of "The Research on Sustainable Tourism in the Light of Its Paradigms"

_sustainability, doi:10.3390/su11205821_

Round 1

Reviewer 1 Report

Good paper with qualitative research 

Author Response

Thank you for the review that gave us lots of useful advice.

It is indeed largely a theoretical article on the concept of sustainability in tourism. The research concept based on the theoretical revision especially - we accentuate this in the Introduction and Methodology.

We will stress the exploratory nature of empirical research (page 1, line 18-19; page 3, line 105; page 13, line 499-502). This will enhance adequacy described methods.
We are satisfied that in your opinion this is very good paper with qualitative research. This kind of research often isn't appreciate.

Reviewer 2 Report

This paper reads more like an essay, not as a scientific paper. It allows some reflection on the different research paradigms.

Results are weak and fail in presenting the Polish perspective of Sustainable tourism and sustainable development.  What are the visible results of the selected sample of articles on sustainable development in Poland. Just adding a long table enumerating the case studies is not enough. More analysis and attention should be paid to this because as the title of the paper suggests, it is the core of the research.

The title is too long and not attractive and I suggest to find a way of reducing it to about its half. Try to use mainly strong substantives and maybe one verb and avoid or restrict the use of "of", "in", "and", "the" etc.

The English needs to be reviewed, as 
I found several mistakes.

Page 8, line 352, the new paragraph starts with «He included economic sciences...». He can not be appropriate, because the text or the previous paragraph is not referring to any person.

Author Response

Thank you for the review that gave us lots of useful advice.

Indeed it is largely a theoretical article on the concept of sustainability in tourism. We make an effort in the theoretical revision and we accentuate this in the Introduction and Methodology. We will stress additionally the exploratory nature of empirical research (page 1, line 18-19; page 3, line 105; page 13, line 499-502).

We want that the article revolve especially around research methodology used by enterprises. We don't analyse the visible results of the selected sample of articles on sustainable development in Poland.
We are aware of the application in the case of Poland is somewhat limited.
The scope of research and analyses could be extend in future. This is very interesting line of study.

We agree with the comment that the title is too long. We will make improvements the title of the article. We hope that his abbreviated version will be better: "The Research On Sustainable Tourism In The Light Of Its Paradigms". The omission of "Polish Perspective" in the title could clearly accent dominant theoretical character of article. Especially that this theoretical study has a more universal character.

We will make English changes required. Thank you for your precise indication; We used phrase: Economic sciences has included among typological sciences (page 8, line 352)

Reviewer 3 Report

It is a theoretical article on the concept of sustainability in tourism. The authors make an effort in the theoretical revision, although the application in the case of Poland is somewhat limited. In any case, it is a theoretical study of interest to researchers in sustainability in tourism when considering theoretical frameworks.

Author Response

Thank you for the review that gave us lots of useful advice.

It is indeed largely a theoretical article on the concept of sustainability in tourism. We make an effort in the theoretical revision and we accentuate this in the Introduction and Methodology. We are aware of the application in the case of Poland is somewhat limited.

We will stress the exploratory nature of empirical research (page 1, line 18-19; page 3, line 105; page 13, line 499-502). We will make improvements the title of the article. We think that his abbreviated version will be better: "The Research On Sustainable Tourism In The Light Of Its Paradigm". The omission of Polish Perspective in the title could clearly accent dominant theoretical character of article. Especially that this theoretical study has a more universal character.